# MTNeuro: A Benchmark for Evaluating Representations of Brain Structure Across Multiple Levels of Abstraction

Jorge Quesada[1]*, Lakshmi Sathidevi[1]*, Ran Liu[1], Nauman Ahad[1], Joy M. Jackson[1],
Mehdi Azabou[1], Jingyun Xiao[1], Christopher Liding[1], Matthew Jin[1], Carolina Urzay[1],
William Gray-Roncal[2], Erik C. Johnson[2,†], Eva L. Dyer[1,†]

1- Georgia Institute of Technology
2 - Johns Hopkins University Applied Physics Laboratory

## Abstract

There are multiple scales of abstraction from which we can describe the same image, depending on whether we are focusing on fine-grained details or a more global attribute of the image. In brain mapping, learning to automatically parse images to build representations of both small-scale features (e.g., the presence of cells or blood vessels) and global properties of an image (e.g., which brain region the image comes from) is a crucial and open challenge. However, most existing datasets and benchmarks for neuroanatomy consider only a single downstream task at a time. To bridge this gap, we introduce a new dataset, annotations, and multiple downstream tasks that provide diverse ways to readout information about brain structure and architecture from the same image. Our multi-task neuroimaging benchmark (`MTNeuro`) is built on volumetric, micrometer-resolution X-ray microtomography images spanning a large thalamocortical section of mouse brain, encompassing multiple cortical and subcortical regions. We generated a number of different prediction challenges and evaluated several supervised and self-supervised models for brain-region prediction and pixel-level semantic segmentation of microstructures. Our experiments not only highlight the rich heterogeneity of this dataset, but also provide insights into how self-supervised approaches can be used to learn representations that capture multiple attributes of a single image and perform well on a variety of downstream tasks. Datasets, code, and pre-trained baseline models are provided at: `https://mtneuro.github.io/`.

## 1 Introduction

Our understanding of our natural surroundings requires multiple levels of perceptual processing: we can recognize a macroscopic object (e.g., a tree), while also identifying finer-grain structures within it (e.g., leaves and branches), and context-relevant features (e.g., leafiness, height, or season). This multi-level perception scheme also translates to the medical image domain: the process of interrogating medical images (either by a human expert or an algorithm) involves combining macrostuctural insights (such as a region of interest) with context-relevant microstructure information and human-interpretable features (e.g., the density of a given cell type in a microscopy image) in order to derive a diagnosis or characterize a target sample.

---

*Equal contribution. Contact authors: (ELD, JQ, LS) {evadyer, jpacora3, lsathidevi3}@gatech.edu; (ECJ) erik.c.johnson@jhuapl.edu; † Both senior authors contributed equally.

35th Conference on Neural Information Processing Systems (NeurIPS 2022) Track on Datasets and Benchmarks.

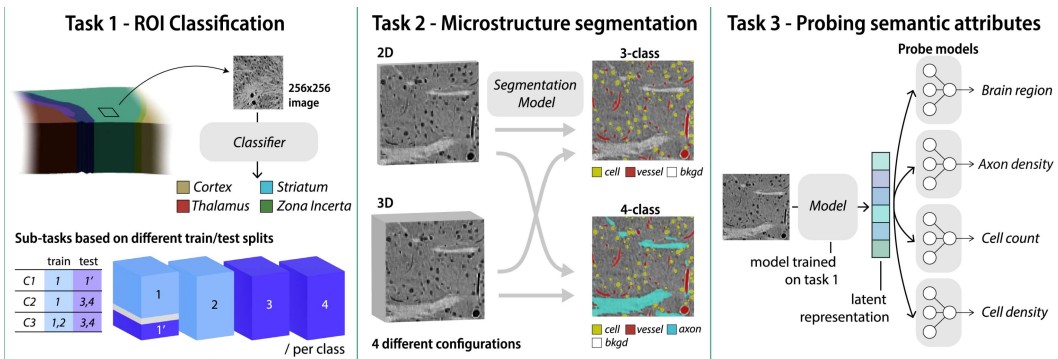

Figure 1: *Overview of* `MTNeuro` *Benchmark*. Task 1: Brain region (macrostructure) classification (3 configurations which vary in data availability and testing schemes); Task 2: Pixel-level microstructure segmentation (4 configurations which vary in sample dimensionality, span and target class count); Task 3: Probing semantic attributes from image-level embeddings obtained in Task 1.

In particular, the ongoing effort to understand the connections and dynamics of the brain involves analyzing both macroscopic level properties such as region-level structures (1; 2) as well as detailed microstructures like the size or shape of a given cell type (3). While significant advances have been achieved in unveiling the properties and structures within the brain through several imaging modalities at different scales (4; 5; 6; 7), most existing neuroimaging benchmarks are designed for evaluation at a single spatial scale, or geared towards a particular downstream task. This can be attributed to several causes, including the prohibitive cost of manual annotation for data spanning multiple scales (8; 9), the associated computational cost of processing multi-scale data, and the fact that neuroimaging technologies have only recently progressed towards pipelines that can capture multi-area volumes at high resolutions.

To fill this gap, we present the `MTNeuro` benchmark (Figure 2): a multi-task, multi-scale benchmark based on a large 3D X-ray microtomography image dataset spanning multiple areas from a mouse brain. Code and access to the data are provided at: `https://mtneuro.github.io/`. We host our dataset in the Brain Observatory Storage Service and Database (BossDB, a specialized interactive database (10)) and provide an integrated dataloader to facilitate transfer experiments and analysis. This benchmark provides a unified framework that allows for evaluating models and representations arising in three distinct tasks:

- **Task 1 - Image-level classification of brain region:** prediction of the brain region (somatosensory cortex, striatum, thalamus, zona incerta) to which a given image belongs.

- **Task 2 - Pixel-level segmentation of microstructures**: prediction of neural microstructures (axons, cell bodies, blood vessels, background) at the pixel-level across the four core brain regions in the dataset.

- **Task 3 - Probing multiple semantic features from learned image-level embeddings**: estimation of semantic (human-interpretable) features (such as the average cell size or axon density) from the representation of a given image, obtained after "freezing" the weights of a trained encoder.

To understand how current models perform on these different tasks, we evaluate a family of different supervised and self-supervised models. Our results in Tasks 1 and 3 highlight a significant generalization gap between self-supervised and supervised approaches, which opens up interesting opportunities for further evaluation and development of self-supervised learning (SSL) methods for these tasks. Through testing across a family of different models across a variety of tasks, our proposed benchmark provides both an exciting platform for evaluating self-supervised learning (SSL) methods, and a rich tool in the effort to extract fundamental insights into brain architecture at both the micro- and macro-scale.

## 2 Background and Related Work

### 2.1 The need for a benchmark in brain mapping and connectomics

Over the past decade, there have been major advances in our ability to resolve fine-scale neuroanatomical structures in the brain. With these advances, we have generated large amounts of brain data that span many spatial scales, and can reveal different features of brain organization. At the nanoscale, electron microscopy has provided detailed wiring diagrams of small portions of cortex (11). At micron scale, microscopy techniques have provided detailed pictures of cytoarchitecture - or how neurons and cells are organized (5). Efforts at even larger scales to capture many brain areas simultaneously, like connectivity atlas and X-ray microtomographic datasets (12), have provided information about the interplay between long-range connections across brain areas and microstructures, such as cell body densities and other morphological features of brain structure.

Accompanying these new tools for data generation have been major advances in machine learning and computational approaches for modeling and analyzing these datasets, for problems such as object detection, segmentation, and classification. While the information provided by these methods are incredibly rich and have a great deal of structure at many scales, any given method is typically tested on an individual challenge at a particular scale. The expense of annotating and proofreading can be considerable, and significant neuroanatomy knowledge is typically required of annotators (9). Moreover, many efforts to provide high quality data, such as (5), have not focused on building ML-oriented benchmarks, but rather on providing reference datasets and resources.

Using machine learning tools to understand these emerging brain datasets at different spatial scales is both a challenge and an increasingly critical need. As a result, large tera- and peta-scale connectomics datasets are being collected using electron microscopy and X-ray microtomography, including data from the entire brain of *Drosophila* (13; 14), large portions of the mouse brain (15; 3) and even a cubic millimeter of human cortex (6). Advances in imaging technologies promise to continually increase the spatial extent, number of species, and number of imaged individuals. These datasets have the micro- or nanoscale resolution and large spatial extents required to resolve subcellular structures (e.g., mitochondria and synapses), microstructures (e.g., glia, neurons, and vasculature), and macrostructure (e.g., brain regions, cortical layer structure, and long-range white matter projections). The multi-scale nature and large size of these datasets requires new ML tools (16), which drives the need for benchmarks that can extract representations of neural structure at different scales.

### 2.2 Existing datasets and benchmarks for resolving brain structure

Due to the large variety of spatial scales, neuroanatomical structure, and imaging modalities, a wide range of segmentation and classification problems have been formulated for neuroimaging data. At the macroscale, there has been a long history of developing benchmarks for different datasets in MRI and related modalities like DTI and fMRI. For example, the BraTS dataset (17) focuses on the MRI of brain tumor and motivated many segmentation works (18; 19; 20; 21). The ADNI (22) and MIRIAD (23) datasets provide MRI-based Alzheimer's disease imaging that focuses on tracking disease progression (24; 25; 26; 27; 28; 29; 30). The UK Biobank (31) offers a huge collection of Brain MRI data from 5,00,000 human participants. The low spatial resolution of MRI and related methods, however, limit the ability to observe microscale structures at the cellular or subcellular level.

For microscale structure, there are fewer benchmarking efforts due to the scale and complexity of the annotation and processing. Example problems include segmentation of synapses in the CREMI challenge (32), which provides high-resolution imaging of synapses in 5 cubic microns of non-isotropic EM with nanometer resolution. Other problem formulations include 2D image segmentation of cell membranes (ISBI 2012 Challenge), with data from (33), and 3D segmentation of cells (34; 35). Specific benchmarks have also been developed for axon instance segmentation (36) and mitochondria segmentation (37). Different forms of microscopy with micrometer resolution, including calcium fluorescence microscopy, are suitable for segmenting cell bodies and extracting functional time-series data, but lack other microstructure information. Benchmarks have also been established for estimation of functional traces from two-photon calcium fluorescence microscopy, including spikefinder (38). These benchmarks have been instrumental in driving progress on specific problems at specific spatial scales. In general, however, most benchmarks at the microscale focus on

relatively small spatial extents, and lack the multi-scale macrostructure, which is also present in the brain. To the best of our knowledge, there are no public microtomography datasets of brain structure with both dense microstructure and macroscale annotations currently available.

Encompassing larger spatial extents, projects such as the BigBrain atlas provides high-resolution sections from the mouse brain with Nissl contrast (39), but the resolution only allows resolution of cells around 20 µm. Large-scale EM datasets are also being used to benchmark performance and segment neurons, augmented with iterative human proofreading (40; 41; 6), which are leading to large segmented datasets with increasingly complex annotation. These data, however, are not suitable for large-scale benchmarking and algorithm development in the general machine learning community due to their size and ongoing refinement. To accelerate progress towards machine learning tools which can operate at multiple levels of spatial abstraction within the same high-resolution dataset, benchmarks are needed which encompass large spatial extents at high resolution. This will enable a broader scientific community to apply state-of-the-art methods for these important applications.

## 3 Dataset and Tasks

### 3.1 Overview of dataset

We build our benchmark on a large open access high-resolution (1.17µm isotropic) 3D X-ray microtomography imaging dataset that provides fine-scale information about brain microstructure as well as diverse regions of interest that give more distinct global attributes (15). The dataset thus contains a uniquely rich set of both *macroscale* (region of interest) and *microscale* (cells, blood vessels, axons) structures that can be interrogated throughout the dataset (42; 16).

The full volumetric dataset provides micron resolution of an intact brain sample totalling $5805 \times 1420 \times 720$ pixels. The dataset spans four regions of interest: somatosensory cortex (CTX), striatum (STR), ventral posterior region of thalamus (VP), or the zona incerta (ZI) (see Figure 2 A-B) and provides pixel-level microstructure (cell, axons, blood vessels, background) and macrostructure labels over 2D slices distributed over the dataset.

To provide more data for training and validation, we expanded the pixel-level microstructure and macrostructure labels provided in the original data resource (15) to larger contiguous volumes in the data. To expand the pixel-level labels, we trained a 4-class Unet model on the sparse 2D annotations and had a trained expert proofread the annotations to create dense pixel-level microstructural labels (see Figure 2B), identifying each point as either part of an axon, cell, blood vessel, or background. The final curated pixel-level labels span 4 ROIs, with each ROI consisting 360 256x256 densely labelled images.

To examine semantic attributes of the different images, in Task 3, we leveraged the dense pixel-level labels to compute a number of semantic features from the reconstructions: (i) the density of blood vessels, (ii) axon density, (iii) the number of cells, (iv) size of cells, and (v) the average inter-cell distance in each slice. These semantic labels provide information into different features of the cytoarchitecture that can be used to interpret the embeddings learned by models tested on this dataset. In addition to these microstructure annotations, we have expanded the macrostructure annotations from the original dataset to include interpolations of the region labels across all 720 slices of size $5805 \times 1420$. From these interpolated sections, we extracted 12 new subvolumes (three for each of the four regions of interest) for examining the generalization of models in Task 1.

### 3.2 Data access

The dataset and all corresponding labels are stored in BossDB (10), the Brain Observatory Storage Service and Database. The dataset project page can be found at: `http://bossdb.org/project/prasad2020`. BossDB is a specialized spatial database for Electron Microscopy and X-Ray Microtomography Datasets, with seamless visualization through Neuroglancer, which enables interactive visualization of large-scale 3D annotated volumes and annotations. All data are available publicly, using public log on credentials (no account creation required). The project page documents project metadata, citation instructions, and the data creators and curators.

For benchmarking, data are accessed through the Python intern API (43). This API allows a remote connection to the BossDB system, including downloads of arbitrary, on-demand 3D cutouts of data,

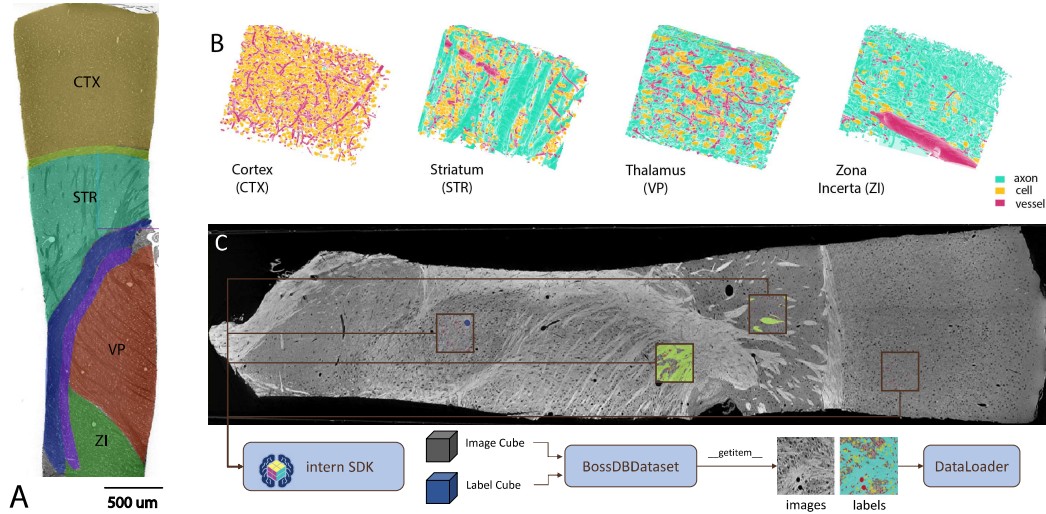

**Figure 2:** *Overview of datasets and annotations used in MTNeuro.* In A we show how the dataset spans multiple brain areas including the somatosensory cortex (CTX), striatum (STR), thalamus (VP), and zona incerta (ZI). Each of these areas contains annotations of pixel-level microstructures like axons, blood vessels, and cells visualized as dense X-ray microCT volumetric imaging data (1.17 micron isotropic resolution), as seen in B. The pipeline and BossDB data access are shown in C: channels are accessed through the Intern API to access cutouts without the need to download the entire dataset. This enables creating a data loader for specific task-relevant cutouts.

including raw images and annotations, without the need to download the entire dataset to disk. To facilitate the use of this API, we provide a Pytorch DataLoader for rapid algorithm development and testing. We also provide sample Jupyter notebooks to demonstrate how the task cutouts can be dowloaded, and saved as Numpy files for development in other frameworks. The tasks are defined with task-specific JSON files specifying the metadata for each task. The use of this dataloader is illustrated in Fig. 2C, where we detail how the dataset can be efficiently accessed through BossDB.

The data are structured into channels. Raw images, macrostructure annotations, and microstructure annotations, each have their own separate channel. This is detailed below.

**Raw images**: All tasks utilize the same raw images, which are single color (grayscale) with 8-bit unsigned integer values. The total raw data volume is $720 \times 1420 \times 5805$ voxels at a resolution of $1.17 \mu m$. This data is available through the raw images channel: `https://api.bossdb.io/v1/mgmt/resources/prasad/prasad2020/image`.

**Macrostructure annotations**: Tasks 1 and 3 utilize dense pixel-level annotations of different brain regions (macrostructure) : CTX (label 0); STR (label 1); VP (label 2); ZI (label 3). At this scale/level, there is an equal number of samples of each class and hence the classes are balanced. This label data is represented with 64-bit unsigned integers and is accessible through the macrostructure annotations channel: `https://api.bossdb.io/v1/mgmt/resources/prasad/prasad_analysis/roi_labels`.

**Microstructure annotations**: Task 2 and Task 3 utilize pixel-level annotations of brain microstructure. Volumes of $256 \times 256 \times 360$ are densely annotated with microstructure labels. The labels are 0: no label (background); 1: blood vessels; 2: cells; 3: mylenated axons. Represented with 64-bit unsigned integers, it is accessible through the microstructure annotations channel: `https://api.bossdb.io/v1/mgmt/resources/prasad/prasad_analysis/pixel_labels`.

A key aspect of our approach is portability of data access and training infrastructure across neuroimaging datasets in BossDB. This allows for easy extension of code and baselines to new volumetric imaging datasets stored in BossDB. These include data from new species, with new microstructure labels (synapses, membranes, mitochondria), and with new macrostructure labels (brain regions, experimental state). By modifying the dataloader JSON, developers can specify different BossDB datasets, spatial regions, and annotation (label) sources. This allows for the flexibility required to support the different tasks in this dataset, and will enable further training and deployment

of these baseline models to new datasets. This is an important contribution of this work towards developing machine learning tools for emerging large-scale neuroimaging datasets.

## 3.3 Tasks

### 3.3.1 Task 1: Image-level classification of brain region-of-interest (ROI)

When analyzing imaging data that spans many different brain regions, one important question is the degree to which global image features correlate with the brain region from which the sample is drawn (16). Thus, we can pose this as a classification problem, where we pull a small patch (or small region) from the data and estimate which of the 4 brain regions the sample was drawn from. Given the abundance of samples available for this task across the entire volume, we can sub-divide this task into three different training schemes (see Figure 1), detailed below. Taken together, these three training schemes allow us to evaluate how different types of models are able to generalize as more data becomes available during training.

**ROI-C1.** For this sub-task, we use the 4 densely-annotated cubes (shown in blue in Figure 1, Task 1), each corresponding to one of the regions of interest (CTX, STR, VP, or ZI). We divide each of the four subvolumes by selecting the first 300 ($256 \times 256$) images slices for training, and the last 50 images for testing, leaving 10-slices between the train and test data to avoid any structural overlap. The resulting overall sample size for this sub-task is thus 1400 images, with 1200 images for the train set and 200 for the test set.

**ROI-C2.** In this sub-task, we evaluate the performance of the models when tested on new areas within the larger 3D context of the dataset. We extract two additional $256 \times 256 \times 360$ cubes per class (shown in Figure 1, Task 1), to serve as additional test data for the models. This sub-task employs 4080 images, with 1200 in the train set, and 2880 in the test set.

**ROI-C3.** In this sub-task, we evaluate the performance of the models when allowed to learn from a larger set of data. We extracted one additional cube per class to serve as an additional source of training data for the models and use the same test set as that employed in ROI-C2. This sub-task employs 5520 images, with 2640 in the train set, and 2880 in the test set.

### 3.3.2 Task 2: Pixel-level segmentation of microstructures

Another important requirement for a comprehensive mapping of brain data is correctly identifying brain microstructures such as cells and blood vessels. In this task, we utilize the dense pixel-level microstructure labels of the volumetric cutouts from 4 brain regions of interest (macrostructure; Cortex, Striatum, VP & ZI), and evaluate different baselines on their prediction accuracy in classifying each pixel from test volumes across the brain regions into appropriate brain microstructure classes (blood vessel, cell, axon & background).

**3-class segmentation task.** We first consider the pixel-level segmentation of images into one of three classes: either cell bodies, blood vessels, or other (background and axons). We use the same train and test split as in ROI-C1 in Task 1 on the main subvolumes that are densely annotated at the pixel-level (300 images for train, 50 for test, with a gap of 10 slices between datasets). This sub-task employs 1200 images for training, and 200 images for testing.

**4-class segmentation task.** In this task, we consider the pixel-level segmentation of images into one of four classes: either cell bodies, blood vessels, background or axons. When we consider dense axonal segmentation, we remove the ZI region from our training and testing set due to the difficulty to reliably segment axons in this subvolume even for human annotators. This sub-task employs 900 images for training, and 150 images for testing.

### 3.3.3 Task 3: Probing multiple semantic features from learned image-level embeddings

In this task, we wanted to explore the possibility of decoding semantic and human-interpretable features from the image-level representations learned in Task 1. If possible, it could provide a way to build interpretable image-level feature maps that contain information about microstructure without needing the expensive pixel-level labels necessary to compute these attributes in most brain mapping settings.

Table 1: *Results on image classification accuracy for brain region prediction (Task 1).*

|  | ROI - C1 | ROI - C2 | ROI - C3 |
|---|---|---|---|
| Supervised | $0.88 \pm 0.03$ | $0.77 \pm 0.03$ | $0.88 \pm 0.02$ |
| Sup w/ Mixup | $0.90 \pm 0.04$ | $0.78 \pm 0.03$ | $0.90 \pm 0.02$ |
| BYOL | $0.88 \pm 0.02$ | $0.76 \pm 0.02$ | $0.97 \pm 0.01$ |
| MYOW | $0.90 \pm 0.02$ | $0.78 \pm 0.05$ | $0.98 \pm 0.01$ |
| MYOW-m | $0.94 \pm 0.02$ | $0.78 \pm 0.03$ | $0.98 \pm 0.01$ |
| PCA | 0.59 | 0.25 | 0.07 |
| NMF | 0.62 | 0.27 | 0.50 |

Specifically, we try to predict the following global properties of an image: (i) blood vessels density, (i) cell count, (iii) average cell size, (iv) axon density, and (v) average distance between cells, all through a simple linear readout. In the case of supervised models, these models are trained to classify images into its respective brain region; However, in the case of SSL methods, where the region-level labels aren't used to guide learning, it may be possible that other global attributes of the images may be encoded in the latent space of the model.

## 4 Results

### 4.1 Task 1: Image-level classification of brain ROI

**Experiment setup.** In this task, we consider the classification of different images into a number of candidate brain areas (CTX, STR, VP and ZI) using representations learned through supervised and self-supervised approaches. All of the models in this task are trained using a Resnet18 encoder (44). We benchmarked two supervised models: the first trained using standard approaches for regularization (20% dropout and weight decay factor of 0.3) and the other trained using mixup (45). Given the underlying shared structure of image samples in volumetric data, we also consider a number of self-supervised learning methods suitable for this task: (i) BYOL (46), (ii) MYOW (47), (iii) and a variant of MYOW that we tested with a single projector and predictor (MYOW-merged, or MYOW-m). For the SSL models, we follow the standard procedure of freezing the network weights after training, and then training a linear layer on top of the representations. This tells us how well the SSL loss captures the classes in the data after only a linear transformation. All models have a latent dimensions of 256. As additional baselines, we also extracted 256-dimensional embeddings from our data using Principal Component Analysis (PCA) and Non-Negative Matrix Factorization (NMF), and trained a linear layer on these representations.

In our experiments, we evaluate all methods across 5 training instances with different random seeds, and report the overall mean accuracy and standard deviation. All models are trained for 100 epochs using an SGD optimizer with a learning rate of 0.03. For more details on our experimental setup and models, see Section 3 in the Appendix.

**Results in classifying brain ROIs.** We report the results of the three subtasks for ROI classification in Table 1. In our first subtask (ROI-C1), we find that many of the SL and SSL models achieve comparable accuracy, with the MYOW-m model achieving the highest accuracy in this limited training regime. In ROI-C2, we test the generalization capabilities of the models trained in ROI-C1 by evaluating how well they performed on subvolumes in other parts of the larger dataset (Table 1, ROI-C2). There is considerable heterogenity across brain regions and thus we can consider this as a form of domain generalization. In this case, we observe a significant decrease in accuracy that can be attributed to the domain shift in data. In ROI-C3, we add another set of training data (see Figure 2) and test on the same volumes as in our last subtask. In this case, we can observe that SSL methods significantly outperform the SL models, with SSL models achieving even higher accuracy than in the small-scale case (97-98%) and supervised models achieving a much more modest improvement (88-90%) with the new training data. We can thus observe a significant gap in SSL over SL models, showing that exposure to additional data (in this case, the additional cube with respect to task ROI-C1) drastically improves the generalization of the SSL models to unobserved data.

### 4.2 Task 2: Pixel-level segmentation of microstructures

In Task 2, we consider four different variants of pixel-level segmentation. The first variant employs 2D models for pixel-level segmentation. The second variant employs 3D models instead. The third

Table 2: *F1 & IoU scores for models trained on the pixel-level segmentation task (Task 2).*

*I. 2D Pixel-level Segmentation*

| Method | Metric | 3-Class | | | | 4-Class | | | | |
|---|---|---|---|---|---|---|---|---|---|---|
| | | Bg + Axons | Vessels | Cells | Avg. | Bg | Vessels | Cells | Axons | Avg. |
| 2D U-Net | F1 | 0.99 | 0.76 | 0.85 | $0.87 \pm 0.012$ | 0.97 | 0.82 | 0.87 | 0.94 | $0.90 \pm 0.003$ |
| 2D U-Net | IoU | 0.98 | 0.64 | 0.75 | $0.79 \pm 0.014$ | 0.89 | 0.70 | 0.77 | 0.60 | $0.74 \pm 0.008$ |
| MA-Net | F1 | 0.99 | 0.79 | 0.87 | $0.88 \pm 0.003$ | 0.97 | 0.83 | 0.87 | 0.94 | $0.90 \pm 0.002$ |
| MA-Net | IoU | 0.98 | 0.68 | 0.78 | $0.81 \pm 0.003$ | 0.89 | 0.71 | 0.78 | 0.76 | $0.78 \pm 0.011$ |
| FPN | F1 | 0.99 | 0.72 | 0.84 | $0.85 \pm 0.01$ | 0.96 | 0.73 | 0.84 | 0.93 | $0.86 \pm 0.004$ |
| FPN | IoU | 0.97 | 0.59 | 0.73 | $0.76 \pm 0.015$ | 0.87 | 0.59 | 0.72 | 0.72 | $0.72 \pm 0.021$ |
| U-Net++ | F1 | 0.99 | 0.79 | 0.87 | $0.89 \pm 0.002$ | 0.97 | 0.81 | 0.85 | 0.93 | $0.89 \pm 0.015$ |
| U-Net++ | IoU | 0.98 | 0.68 | 0.78 | $0.81 \pm 0.002$ | 0.88 | 0.68 | 0.75 | 0.73 | $0.76 \pm 0.036$ |
| PAN | F1 | 0.98 | 0.60 | 0.80 | $0.79 \pm 0.035$ | 0.95 | 0.69 | 0.80 | 0.93 | $0.84 \pm 0.007$ |
| PAN | IoU | 0.96 | 0.46 | 0.66 | $0.69 \pm 0.039$ | 0.85 | 0.53 | 0.67 | 0.76 | $0.70 \pm 0.014$ |
| PSPNet | F1 | 0.97 | 0.48 | 0.74 | $0.73 \pm 0.013$ | 0.94 | 0.54 | 0.71 | 0.91 | $0.78 \pm 0.012$ |
| PSPNet | IoU | 0.94 | 0.39 | 0.61 | $0.65 \pm 0.043$ | 0.82 | 0.38 | 0.55 | 0.74 | $0.62 \pm 0.015$ |

*II. 3D Pixel-level Segmentation*

| Method | Metric | 3-Class | | | | 4-Class | | | | |
|---|---|---|---|---|---|---|---|---|---|---|
| | | Bg + Axons | Vessels | Cells | Avg. | Bg | Vessels | Cells | Axons | Avg. |
| 3D U-Net | F1 | 0.99 | 0.77 | 0.87 | $0.88 \pm 0.006$ | 0.93 | 0.76 | 0.80 | 0.87 | $0.84 \pm 0.032$ |
| 3D U-Net | IoU | 0.98 | 0.65 | 0.76 | $0.80 \pm 0.007$ | 0.81 | 0.62 | 0.67 | 0.50 | $0.65 \pm 0.045$ |
| VNetLight | F1 | 0.99 | 0.75 | 0.83 | $0.85 \pm 0.012$ | 0.90 | 0.65 | 0.73 | 0.76 | $0.76 \pm 0.063$ |
| VNetLight | IoU | 0.97 | 0.61 | 0.70 | $0.76 \pm 0.013$ | 0.78 | 0.46 | 0.58 | 0.43 | $0.56 \pm 0.061$ |
| HighResNet | F1 | 0.99 | 0.74 | 0.84 | $0.85 \pm 0.019$ | 0.89 | 0.51 | 0.73 | 0.77 | $0.72 \pm 0.083$ |
| HighResNet | IoU | 0.97 | 0.61 | 0.72 | $0.77 \pm 0.026$ | 0.73 | 0.35 | 0.58 | 0.42 | $0.52 \pm 0.075$ |

variant is a 4-class setting where ZI is removed from the brain regions involved (as axons in this region are difficult to distinguish accurately for even human annotators). The fourth variant is a 3-class setting where all 4 brain regions (Cortex, Striatum, VP and ZI) are utilized but only 3 classes are considered (blood vessels, cells, background+axons; avoids axon segmentation).

**Experiment setup.** In our experiments, we perform pixel-level segmentation using a selected set of 2D and 3D models. Each model is put through a separate hyper-parameter tuning process for finding an optimal learning rate and batch size by training on the train set and evaluating on the validation split. Each model is trained for 20 epochs with its optimal learning rate and batch size and evaluated across 5 training instances (each with its own random seed). The class-wise F1-score, the class-wise IoU, and the overall mean and standard deviation of both metrics are reported for each model (Table 2). We do not report accuracy because for this particular task of pixel-level segmentation, it does not aptly represent the model performance due to class imbalance (see Appendix Section 4.2.3 for breakdown of different classes in the training and test sets).

The models we used for the 2D segmentation task are the standard 2D U-Net model (48; 49) and selected models from the 'segmentation_models.pytorch' library (50): MAnet (51), FPN (52), U-Net++ (53), PAN (54) and PSPNet (55). The models we used for the 3D segmentation task are the standard 3D U-Net model (48) and selected models from 'MedicalZooPytorch' (56): VNetLight (57; 56) and HighResNet (58). For more details refer Section 4 in the Appendix.

**Pixel-level segmentation in 2D.** The results from the selected models on our 2D pixel-level segmentation task are tabulated in Part I of Table 2 and visualized in Figure 3. The individual slices are the input to the models and they are fed in a batched manner during training. For training and evaluation, we consider both the 3-class and 4-class settings and we use EfficientNet-b7 encoder for all the models as it was seen to give the best performance among the 25 encoders that were attempted. From our results we see that MA-Net performed the best overall among the 2D models with an average IoU of 0.78 followed by U-Net++ with an average IoU of 0.76 in the 4-class setting (Table 2). As can be seen from the class-wise IoU and class-wise F1-scores, for most of the best performing models, the most challenging components to differentiate are cells and blood vessels, which are also difficult for human annotators to identify from 2D slices without further 3D context. Also, we can note, the average IoU (across classes and models) increases from 0.72 to 0.79 upon moving from 4-class to 3-class setting. This indicates an expected performance improvement as there are more slices (more data) and fewer classes (easier) in the 3-class setting.

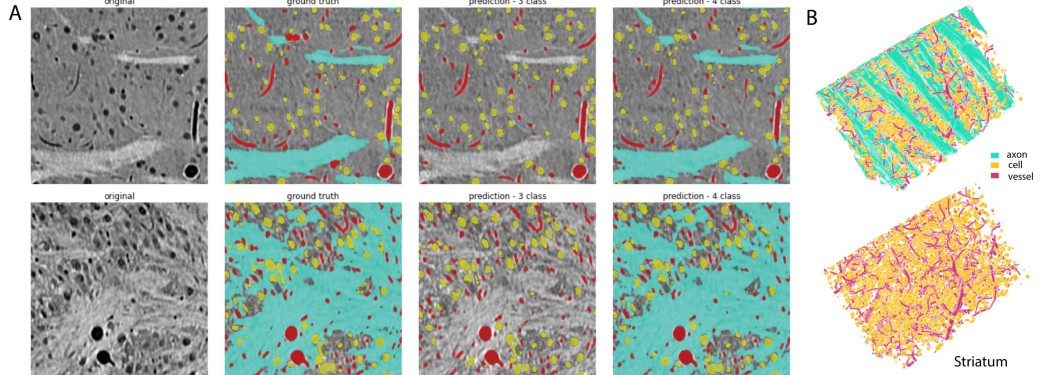

Figure 3: *Visualization of predictions from pixel-level segmentation task (Task 2).* From left to right: original image, ground truth overlaid, prediction from U-Net model in the 3-class and 4-class settings (top row is Striatum, bottom is VP). Axons are visualized in cyan, blood vessels in red, and cell bodies are in yellow. B) 3D reconstruction from the predictions for Striatum for the 4-class (top) and 3-class (bottom) settings.

**Pixel-level segmentation in 3D.** We provide the same breakdown and accuracy measures as in the 2D case, for this 3D case (Part II of Table 2). For providing 3D input to the models, we pass in consecutive slices (8 slices) as a single subvolume and build a prediction over all slices jointly in 3D. The U-Net model performed best with 0.80 IoU (3-class) which is only competent with the best 2D models. This would be because we define the subvolumes in a non-overlapping manner, so even though we have improved 3D context, the model also sees fewer samples/inputs. Due to the same reason, moving from 3-class to 4-class setting, we can note a drop in performance since there are fewer samples. Larger sample counts / allowing overlap could enable 3D models to perform better.

### 4.3    Task 3: Probing multiple semantic features from learned image-level embeddings

In Task 3, we explore the prediction of different semantic attributes (e.g., density of blood vessels or cells) from the latent space learned by the models trained in Task 1.

To do this, we leveraged the high-quality dense microstructure annotations to extract information about the attributes of each image that included: the proportion of pixels that are either blood vessels or axons, the cell count and size, and average distance between cells and their nearest neighbor. Further details on the experimental setup for this task can be found in the Appendix in Section 5.

We use the trained models from Task 1, freeze their weights, and compute the representations of all of the annotated images used in ROI-C1 (both train and test, 360 slices). From these latent

Table 3: *Task 3: $R^2$ scores on multi-task feature readout for supervised and SSL models trained in Task 1.*

| *I. Linear Readouts from Models Trained on a Single Subvolume (ROI-C1)* | | | | | |
|---|---|---|---|---|---|
| Methods | Vessels | Axons | Cell Count | Cell Size | Dist (k=1) |
| Supervised | $0.77 \pm 0.06$ | $0.94 \pm 0.01$ | $0.67 \pm 0.06$ | $0.61 \pm 0.05$ | $0.48 \pm 0.05$ |
| Sup w/ Mixup | $0.82 \pm 0.02$ | $0.95 \pm 0.00$ | $0.71 \pm 0.02$ | $0.67 \pm 0.03$ | $0.47 \pm 0.02$ |
| BYOL | $0.85 \pm 0.01$ | $0.94 \pm 0.01$ | $0.75 \pm 0.01$ | $0.69 \pm 0.01$ | $0.49 \pm 0.01$ |
| MYOW | $0.85 \pm 0.01$ | $0.94 \pm 0.01$ | $0.74 \pm 0.01$ | $0.69 \pm 0.01$ | $0.50 \pm 0.02$ |
| MYOW-m | $0.87 \pm 0.01$ | $0.95 \pm 0.01$ | $0.77 \pm 0.01$ | $0.69 \pm 0.01$ | $0.51 \pm 0.01$ |
| PCA | 0.75 | 0.82 | 0.55 | 0.47 | 0.31 |
| NMF | 0.81 | 0.85 | 0.59 | 0.55 | 0.34 |
| *II. Linear Readouts from Models Trained on Two Subvolumes (ROI-C3)* | | | | | |
| Methods | Vessels | Axons | Cell Count | Cell Size | Dist (k=1) |
| Supervised | $0.79 \pm 0.02$ | $0.94 \pm 0.02$ | $0.73 \pm 0.02$ | $0.63 \pm 0.04$ | $0.49 \pm 0.02$ |
| Sup w/ Mixup | $0.75 \pm 0.04$ | $0.88 \pm 0.04$ | $0.64 \pm 0.04$ | $0.54 \pm 0.07$ | $0.37 \pm 0.05$ |
| BYOL | $0.88 \pm 0.00$ | $0.96 \pm 0.00$ | $0.79 \pm 0.00$ | $0.73 \pm 0.01$ | $0.53 \pm 0.02$ |
| MYOW | $0.88 \pm 0.01$ | $0.96 \pm 0.00$ | $0.79 \pm 0.01$ | $0.72 \pm 0.01$ | $0.52 \pm 0.01$ |
| MYOW-m | $0.87 \pm 0.01$ | $0.96 \pm 0.01$ | $0.78 \pm 0.01$ | $0.72 \pm 0.01$ | $0.53 \pm 0.01$ |
| PCA | 0.75 | 0.82 | 0.53 | 0.46 | 0.29 |
| NMF | 0.75 | 0.83 | 0.56 | 0.49 | 0.31 |

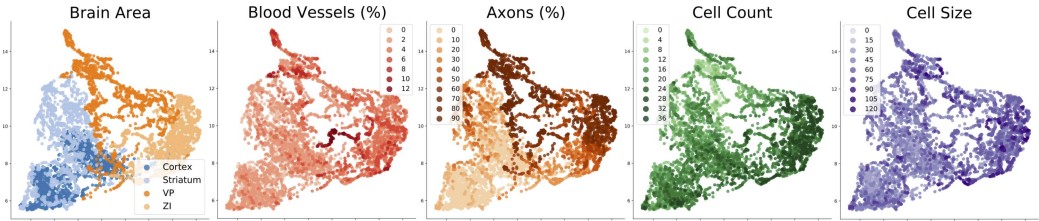

Figure 4: *Visualization of the representations learned by MYOW-m in Task 1 (ROI-C1) with global semantic attributes for each image visualized as different colors.* From left to right, latents are colored by brain area (class), % blood vessels, % axons, cell count, and cell size.

features, we fit a simple linear regression model on these representations to predict each of the desired semantic attributes. We have limited data for this task and thus report the $R^2$ on all of the latents (no train and test split in this case).

We report the $R^2$ values in Table 3 for all 360 slices in the densely annotated volume used in Task 1 and 2 for the models trained in ROI-C1 (Table 3 I) and on more data in ROI-C3 (Table 3 II). In both cases, we find that the SSL models significantly outperform the rest of the baseline models. The gap is more pronounced in the two volume training condition, highlighting the generalization difference between supervised and SSL approaches (59) in providing good representations for a wide range of tasks. In Figure 4, we project the MYOW-m (ROI-C1) embeddings using UMAP (60), and overlay the semantic features.

## 5    Discussion

In this paper, we introduced a multi-task benchmark for analysis of brain structure from high-resolution neuroimaging data spanning many brain areas. In addition, we also built general infrastructure for training models from datasets in the BossDB framework, like dataloaders which can be adapted to different datasets. This will expand the use of large-scale volumetric neuroimaging data for machine learning tool development.

**Limitations and future work.** In neuroscience, often obtaining high quality labeled data (especially for dense segmentation tasks like those provided in Task 2), is very costly (61). The intensive nature of manual annotation and proofreading data thus limits the amount of labeled and annotated data that we provide to train models on, or the amount of distributional shift that can be assessed. While this is a limitation of the work, it also is an accurate reflection of the challenges faced in the field, and thus requires more label-efficient approaches for learning like the SSL methods we highlight. Moving forward, we hope to leverage the models tested in this work to generate even more high quality annotated data to further improve model performance and segmentation.

When designing our current benchmark, we focused on building a multi-scale challenge where variability was due to changes in brain structure and not different preparations or imaging parameters. Therefore we focused on a single animal where we have a large intact brain volume that spans many heterogeneous brain regions. While this may limit the generalization of the models to new samples or datasets, it also addresses the heterogeneous nature of different brain regions that is often overlooked. Our results show that even within a single brain, there is rich heterogeneity across different brain regions that makes it difficult for some models to generalize. This also reveals important generalization gaps between SL and SSL models.

In the future, we hope to expand this effort to learn models of brain structure from other non-convolutional architectures (e.g., point cloud-based models of neural structure (62)), and deploy our tools on new multi-scale brain datasets, perhaps using new lightsheet (63) or whole-brain scaling (64; 65) techniques.

**Broader impacts.** The high heterogeneity of brain data, coupled with the variability in the scale and nature of the tasks presented in this benchmark make it a challenging and useful resource for the broader ML community. Furthermore, we hope the provided dataset and tasks, as well as the supporting codebase and BossDB infrastructure will help accelerate development of ML techniques for emerging, high-resolution neuroimaging datasets being collected in the broader community.

## Acknowledgements

This project was supported by NSF award IIS-2039741, NIH award 1R01EB029852-01, NIH award R01MH126684, NIH award R24MH114785 in addition to generous gifts from the Alfred Sloan Foundation, the McKnight Foundation, and the CIFAR Azrieli Global Scholars Program.

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
