# OpenReview forum: "MTNeuro:  A Benchmark for Evaluating Representations of Brain Structure Across Multiple Levels of Abstraction"
_NeurIPS.cc/2022/Track/Datasets_and_Benchmarks — NeurIPS 2022 Datasets and Benchmarks _

### Official Review · Reviewer_aUux · 2022-07-25
**brain structure extraction benchmark**

**Rating:** 8
**Confidence:** 3
**Clarity:** The paper is clearly written and well…

**Strengths:**

This paper provided a well-organized 3D X-ray microtomography image dataset from multiple cortical and subcortical areas of the mouse brain. This dataset has labels at different scales, from cells, microstructure to brain areas. The authors took advantage of this dataset and carefully structured a variety of tasks and sub-tasks to train neural networks. This dataset has the potential to benchmark different types of image segmentation and classification models, as proposed in this paper. Because of the richness of the dataset, it is relevant to the broader research community. The authors also provided a website which has access to a Github repo including models and a database.

**Weaknesses:**

One potential concern is the generalizability of this benchmark since the dataset is from mouse brain. Because of the unique statistical property of the X-ray brain imaging data, the results of the trained models may not generalize to images from other imaging techniques or organs.


**Additional Feedback:**

None.

**Correctness:**

The dataset contain ground truth labels and the model prediction is evaluated against the labels for different levels of tasks.

**Documentation:**

The documentation of the dataset is very clear, with proper figures and illustrations.
Github repo also contain code for models and tutorials.

**Ethics:**

No.

**Relation To Prior Work:**

This paper reviewed relevant existing neuroimaging datasets and benchmarks for the different levels of tasks they would like to test. The major contribution of this benchmark is that a variety of tasks at different spatial scales can be constructed and tested with this dataset.

**Summary And Contributions:**

Deep learning algorithms have been widely applied to brain structure identification and brain mapping, especially for research and diagnosis purposes. Most of current models and datasets are designed for a single downstream task. It would be useful to develop algorithms that can extract different levels of information, ranging from fine-grained details to global structure all at once, from the same image. This submission aims to establish a benchmark dataset and pipeline for such purpose and defined three example downstream tasks: Image-level classification of brain area; Pixel-level segmentation of microstructures; Multi-task decoding from frozen representations. They hosted a large 3D X-ray microtomography image dataset from mouse brain in the Brain Observatory Storage Service and Database and provided an integrated data loader. They evaluated the performance of a variety of supervised and self-supervised models on those three tasks with this dataset and found that self-supervised algorithms are particularly good in learning a good representation for diverse downstream tasks.

---

> ### Author Response · Authors · 2022-08-11
> **Response to reviewer aUux**
>
> We thank the reviewer for the thoughtful comments and feedback. We appreciate your enthusiasm for the work!
>
> ***1. “One potential concern is the generalizability of this benchmark since the dataset is from mouse brain. Because of the unique statistical property of the X-ray brain imaging data, the results of the trained models may not generalize to images from other imaging techniques or organs.”***
>
> While it is true that the unique statistics of X-ray brain imaging may make it hard to translate features learned from this dataset to other domains, we note that the variability in the scale and nature of the tasks provides a set of unique challenges that we believe are of great interest to the ML and neuroscience communities. Furthermore, we note that this benchmark will enable development of models and representations that could be easily adapted to similar volumetric data on the human brain, if and when it becomes available.
>
> In terms of generalization to other organs or tissue types, we believe that a unique attribute of our current benchmark is its inherent diversity in tissue types and microstructures. Our benchmark provides tasks that span many regions, requires generalization across very diverse tissue types (see images of cortex vs striatum for instance), and provides a good testbed for algorithms that need to transfer in cases where the overall structure and distribution of cells and other microstructures are extremely different. We imagine that computer scientists that want to test their algorithms on new medical imaging datasets, could use a number of different tasks that we provide in our benchmark.
>
> Thanks again for your detailed review and feedback! Let us know if you have further questions.

---

### Official Review · Reviewer_RzzT · 2022-07-26
**Interesting paper that introduces a new benchmark for brain structure**

**Rating:** 6
**Confidence:** 2

**Strengths:**

- The paper introduces a new dataset with multiple tasks at different level of abstraction. A dataset that is designed for different tasks are not frequently seen.

- The API for downloading the data seems well maintained (at least for pytorch users). The authors have gone extra miles to offer usability convenience for computer scientists.

**Weaknesses:**

- Unfortunately, some very basic details are missing, e.g., number of samples, which is a critical factor to evaluate the dataset. Also, the authors direct the readers to their portal for more information (instead of presenting it in the papers), which is arguably inappropriate for a dataset paper.

- It's unclear how these tasks are intercorrelated. For example, won't a successful segmentation mostly indicate the ROI prediction? Also, is Task 3 basically supervised learning with an additional set of labels?

**Additional Feedback:**

- I understand that there is way more information on the website and 9-page might not even be enough to move all the information here, but as a dataset paper, directly directing the readers to the website without offering at least some critical details might not be good. For example, I believe the number of samples (for each task) is a critical factor, but I cannot find it within the manuscript.

- The authors might want to provide line numbers in the submission for more efficient discussions.

**Clarity:**

- Mostly well-written.

- Not sure what "readout" means exactly in the manuscript. It will probably be better to define it with machine learning terms clearly.

- Similarly, the word choice "frozen" seems lost in the context. Fixed weights might be able to deliver the message more straightforwardly.

**Correctness:**

The details are on their website, so unable to evaluate the data correctness within the manuscript.

**Documentation:**

The authors direct the readers to their websites for details, which might be OK, but probably inappropriate for a paper that is primarily about the dataset.

**Ethics:**

none noted.

**Relation To Prior Work:**

Well discussed.

**Summary And Contributions:**

The paper mainly introduces a new benchmark datasets for the community to study brain structure. The dataset comes with multiple different tasks, at different levels of abstraction. The dataset seems offering a new opportunity of studies. The paper also introduces some initial experiments for the study.

---

> ### Author Response · Authors · 2022-08-11
> **Response to reviewer RzzT**
>
> We thank the reviewer for the thoughtful comments and feedback. In what follows, we provide (i) responses to your main comments below and (ii) pointers to the revised text where we have revisions added (major changes and added text are highlighted in Red).
>
> ***1. “It's unclear how these tasks are intercorrelated. For example, won't a successful segmentation mostly indicate the ROI prediction? Also, is Task 3 basically supervised learning with an additional set of labels?”***
>
> Tasks 1 and 2 are independent from each other, and good performance in one does not guarantee good performance on the other, since each particular task may not make use of the same features. Task 3 is in fact supervised learning (linear regression) using the models trained on task 1, and trying to predict semantic features. We note that future studies could benefit by looking at the relevance of semantic features for a given task.
>
> ***2. “Not sure what "readout" means exactly in the manuscript. It will probably be better to define it with machine learning terms clearly.”***
>
> ***“Similarly, the word choice "frozen" seems lost in the context. Fixed weights might be able to deliver the message more straightforwardly.”***
>
> We have contextualized the term ‘readout’ in the text (lines 338-339) and changed the term ‘frozen’ to ‘fixed weights’ to improve readability.
>
> ***3. “I understand that there is way more information on the website and 9-page might not even be enough to move all the information here, but as a dataset paper, directly directing the readers to the website without offering at least some critical details might not be good. For example, I believe the number of samples (for each task) is a critical factor, but I cannot find it within the manuscript."***
>
> ***"The authors might want to provide line numbers in the submission for more efficient discussions."***
>
> The number of samples varies dependending on whether processing happens at the 2D or 3D level and we report them to some extent in the tasks descriptions, but we agree that these details should be as explicit as possible. To add more clarity regarding the dataset, dataset specifications and task specifications, we have improved the text with the dataset overview section (section 3.1). Also we have added sample size, class and other details in lines 188, 193, 216, 221, 227 & 234.
>
> Thanks again for your detailed review and feedback! Let us know if you have further questions.

---

> > ### Comment · Reviewer_RzzT · 2022-08-24
> > **Response to the Rebuttal**
> >
> > Thanks for your update, unfortuantely, I do not find the response to the first question very convincing. "each particular task may not make use of the same features" is only a generic remark that can use to respond to the questions of any two tasks, but I'm asking in particular for two tasks that are fairly related.
> >
> > Nonetheless, since I already gave a rating that favors the paper, I'm reluctant to further improve it. Thanks for the response.

---

### Official Review · Reviewer_JzxN · 2022-07-27
**MTNeuro: A Benchmark for Evaluating Representations of Brain Structure Across Multiple Levels of Abstraction Review**

**Rating:** 6
**Confidence:** 2
**Correctness:** Yes

**Strengths:**

The paper is proposing a dataset that analyzes the differences in fine-grain data collection. This is useful for creating more robust models.

The experimental process and data are well documented.

The results show high accuracy, meaning the work is promising.

**Weaknesses:**

The differences between tasks is not extremely clear for someone who is not knowledgeable about neurology.

Was unable to find the amount of subjects in the experiment

Figures A, B, and C are not very clear and need to be enlarged

Would be great if the authors can briefly explain their choice of models as well as why might SSL be better than SL.

**Additional Feedback:**

N/A

**Clarity:**

There are a few points of clarification I would like to request from the authors:

Are the tasks in increasing order of complexity?

How many subjects were used in the experiment?

Has something similar been tried with human subjects/is that the future direction?

Furthermore, as a preferential detail, the figures could be enlarged and placed on the top and bottom of pages.

**Documentation:**

See second point in Weaknesses

**Relation To Prior Work:**

Compared with existing datasets, OLIVES contains a comprehensive set of modalities and is large enough in volume to be leveraged by ML algorithms. According to the paper this is currently the largest and most diverse dataset of its kind.



**Summary And Contributions:**

The paper builds on an existing database for brain structures by labeling image data on macroscopic and microscopic scales. Using this data, the authors propose classification tasks that might be helpful and benchmark several learning algorithms for this task. The goal is to use high quality neuroimaging of the whole brain for task classification in mice. The dataset is tested using various models on multiple different tasks. The purpose is to provide insights on the functions of different levels of the brain. The Authors provide public access to the dataset, a clear website, and well-documented code for integrating their work into future works.

---

> ### Author Response · Authors · 2022-08-11
> **Response to reviewer JzxN**
>
>
> We thank the reviewer for the thoughtful comments and feedback. In what follows, we provide (i) responses to your main comments below and (ii) pointers to the revised text where we have revisions added (major changes and added text are highlighted in Red).
>
> ***1. “The differences between tasks is not extremely clear for someone who is not knowledgeable about neurology.”***
>
> We had initially provided the details of the tasks in section 3.2 and 3.3 respectively. Noting the need to be more clear, we have added more details regarding the tasks in lines 150-163 (section 3.1), 186-195 (section 3.2), and in lines 216, 221 and 227 & 234 (section 3.3).
>
> ***2. “Would be great if the authors can briefly explain their choice of models as well as why SSL might be better than SL.”***
>
> For tasks 1 and 3, we tried covering as many representative frameworks as possible, including traditional dimensionality reduction (PCA, NMF), and standard supervised networks. We also chose to consider SSL approaches given that they have been shown to perform well under limited data constraints and are highly label efficient, which is a critically desirable feature in a domain where annotation cost is usually a limiting factor, such as neuroanatomy. There is some existing literature that explores SSL generalizability over SL [1], which we will include to further complement our results. For task 2, we chose our models from a family of models made available through the ‘segmentation_models.pytorch’ [2] and MedicalZooPytorch repositories [3].
>
> ***3. “Compared with existing datasets, OLIVES contains a comprehensive set of modalities and is large enough in volume to be leveraged by ML algorithms. According to the paper this is currently the largest and most diverse dataset of its kind.”***
>
> We are happy to add more references to the related work (can you point us to OLIVES? We found a paper using OCT for “eye semantics” but are not sure it's the one you are referencing); However, we want to stress that the focus of the work was towards building a high-quality, high-resolution and multi-scale set of data and labels that can be used to build models to accelerate brain mapping and not towards multi-modal analysis. While we foresee the models and approaches being used in other modalities, this X-ray dataset provides a unique opportunity to access diverse brain structures and tissues in 3D and within an intact brain.
>
> ***4. “Was unable to find the amount of subjects in the experiment”***
>
>  The dataset comes from a large cutout of a single mouse brain. We have updated the manuscript to clarify this detail in lines 42 and 136.
>
> ***5. “Has something similar been tried with human subjects/is that the future direction?”***
>
> There are human post-mortem x-ray datasets at this resolution; however, to the best of our knowledge, this data has been interrogated at the single-task level (usually through segmentation for the study of cytoarchitecture), as the cost of labeling these kind of data is usually a limiting factor. We note that our benchmark will enable development of multi-task models and representations that could be transferable to similar volumetric data on the human brain.
>
> ***6. “Are the tasks in increasing order of complexity?”***
>
> The tasks operate on different levels of abstraction: image-level (task 1), pixel-level (task 2) and semantic level (task 3). The abstraction level does not strictly correlate with the complexity of the tasks (the numbering choice was arbitrary).
>
> **7. Figure clarity**
>
> Thank you for your comments on the visibility of figure elements. We will revise the figures with your feedback in mind.
>
> Thanks again for your detailed review and feedback! Let us know if you have further questions
>
> ---
> **References:**
>
> [1] Atharva Tendle, Mohammad Rashedul Hasan "A study of the generalizability of self-supervised representations". Machine Learning with Applications, Volume 6 (2021).
>
> [2] Pavel Yakubovskiy, “Segmentation models pytorch,” https://github.com/qubvel/segmentation_models.pytorch, 2020.
>
> [3] Adaloglou Nikolaos, “Deep learning in medical image analysis: a comparative analysis of multi-modal brain-mri segmentation with 3d deep neural networks,” M.S. thesis, University of Patras, 2019, https://github.com/black0017/MedicalZooPytorch

---

### Official Review · Reviewer_ncab · 2022-07-27
**NeurIPS 2022 Track Datasets and Benchmarks Paper449 Reviewer ncab**

**Rating:** 7
**Confidence:** 4
**Clarity:** Yes.

**Strengths:**

This sort of data set and benchmarking is badly needed in biomedical image processing. The use of annotations at multiple scales is very much appreciated.

**Weaknesses:**

* The authors should not entirely avoid discussing the limitations of their work simply because it's the data set and benchmark track. Datasets and benchmarks absolutely also have limitations.
* All of the data originates from a single animal and a single preparation procedure. While not "making perfect the enemy of the good", is there no other public data set of microtomography that the authors could compare against for even the ROI categorization task?
* In the appendix, for tasks 1 and 2 apparently one or more hyperparameter sets were varied for each task but only the best performing was reported. For a benchmark I would expect at the bare minimum the appendix to detail for each set which hyperparameter combination was the highest performing for each task; I would also personally say that the results of all should be reported.

**Additional Feedback:**

None.

**Correctness:**

Yes, though in a perfect world the final benchmark computations would have been performed on a set that had been held out during hyperparameter optimization. If that was indeed done, it is not explained clearly enough here.

**Documentation:**

No discussion of maintenance, and/or ethical and responsible use.

**Ethics:**

No ethical concerns.

**Relation To Prior Work:**

The authors discuss in general how this work differs from the rest of the field, but do not sufficiently clarify how it differs from their own previous work (Reference 12).

**Summary And Contributions:**

The authors provide a microtomography data set containing ROI annotations for a very large brain volume, as well as dense pixel annotations for several subvolumes. They set benchmarks for several tasks using this data set - class prediction, semantic segmentation, as well as testing the suitability of their class prediction models for multi-task problems.

---

> ### Author Response · Authors · 2022-08-11
> **Response to reviewer ncab (1/2)**
>
>
> We thank the reviewer for the thoughtful comments and feedback. In what follows, we provide (i) responses to your main comments below and (ii) pointers to the revised text where we have revisions added (major changes and added text are highlighted in Red).
>
> ***1. “All of the data originates from a single animal and a single preparation procedure. While not "making perfect the enemy of the good", is there no other public data set of microtomography that the authors could compare against for even the ROI categorization task?”***
>
> There are a number of X-ray or brain imaging datasets available that either span a single brain area [1,2] or lack resolution of the microstructures [3,4] that are the focus of our work. Moreover, there is high variability across imaging sessions and parameters that introduce additional sources of noise into the data and make it harder to assess the finer-scale differences across brain areas. Thus, we believe adding additional datasets for comparison on single tasks might be detrimental to conveying the overall message/contribution of the paper, i.e the multi-task and multi-scale nature of this benchmark. Also, to the best of our knowledge, there isn’t an existing publicly available microtomography dataset of the brain having both dense microstructure annotations and macroscale labels that is currently available. However, we do agree with the reviewer that variability across animals and across imaging parameters can be significant and may also provide additional challenges when aiming to build a unified model across animals. We detail these challenges and drawbacks in the discussion section in Lines 356-374.
>
> ***2. “The authors discuss in general how this work differs from the rest of the field, but do not sufficiently clarify how it differs from their own previous work (Reference 12).”***
>
> This benchmark paper adds additional pixel-level annotations on the existing dataset both at the micro and macro-structural levels, new features extracted from the fine-scale microstructure segmentations, as well three evaluation tasks at different levels of abstraction. In particular, the original paper provided 11 microstructure-annotated 2D slices for each of the 4 subvolumes corresponding to the interest regions. In this work, we provide full microstructure annotations for each subvolume, therefore adding 349 annotated slices for each of the 4 subvolumes. Furthermore, the original paper provided 8 region-level slices as macrostructural annotations. We have annotated the regions in between these slices, resulting in 342 new region-level slice annotations. We have added Lines 150 - 163 (section 3.1) in the revised paper to convey with more clarity how our current work differs from our previous work.
>
> ***3. “No discussion of maintenance, and/or ethical and responsible use.”***
>
> We appreciate this comment from the reviewer on these key points which were not adequately addressed. The dataset, data access API, and dataloaders will be maintained by the BossDB team, which is tasked as the BRAIN Initiative archive of record for nanoscale connectomics datasets. The team is developing standards in accordance with FAIR principles (https://www.go-fair.org/fair-principles/) to ensure permanent identifiers and broad accessibility. Data and data access APIs will be reviewed semi-annually and community issues can be addressed in standard two-week sprint cycles by the development team. The baseline code is available open source, and community forks and pull requests will be welcome, to be reviewed by the repository maintainers. We anticipate the use of the baselines and infrastructure for further semantic segmentation and challenges on BossDB. As improvements are made to the baseline codebase for future challenges, changes will be pushed to the MTNeuro repository as a new versioned release. We understand the value of a clear discussion of maintenance, and ethics so we have included a section that discusses maintenance in the appendix for the revised paper in lines 670 to 684 (Appendix Section 2).
>
> ***4. “In the appendix, for tasks 1 and 2 apparently one or more hyperparameter sets were varied for each task but only the best performing was reported. For a benchmark I would expect at the bare minimum the appendix to detail for each set which hyperparameter combination was the highest performing for each task; I would also personally say that the results of all should be reported.”***
>
> Thank you for pointing this out, we are currently compiling the optimal hyperparameters used for obtaining the performance detailed in Table 2 and will be adding those details to the Appendix.
>
> (continued...)

---

> > ### Author Response · Authors · 2022-08-11
> > **Response to reviewer ncab (2/2)**
> >
> > **References:**
> >
> > [1] Dyer, Eva L. et al. "Quantifying Mesoscale Neuroanatomy Using X-Ray Microtomography". eNeuro 4. 5(2017).
> >
> > [2] Kuan, A.T., Phelps, J.S., Thomas, L.A. et al. Dense neuronal reconstruction through X-ray holographic nano-tomography. Nat Neurosci 23, 1637–1643 (2020). https://doi.org/10.1038/s41593-020-0704-9.
> >
> > [3] Foxley S, Sampathkumar V, De Andrade V, Trinkle S, Sorokina A, Norwood K, La Riviere P, Kasthuri N. Multi-modal imaging of a single mouse brain over five orders of magnitude of resolution. Neuroimage. 2021 Sep;238:118250. doi: 10.1016/j.neuroimage.2021.118250. Epub 2021 Jun 9. PMID: 34116154; PMCID: PMC8388011.
> >
> > [4] Griffin Rodgers, et al. "Virtual histology of an entire mouse brain from formalin fixation to paraffin embedding. Part 1: Data acquisition, anatomical feature segmentation, tracking global volume and density changes". Journal of Neuroscience Methods 364. (2021): 109354

---

> > ### Comment · Reviewer_ncab · 2022-08-18
> > **Review of revised manuscript**
> >
> > In general, the manuscript is much improved, though the authors and I will have to agree to disagree on the feasibility and importance of finding another data set for testing in Task 1, since I think ultimately the utility of this particular data set is substantially reduced in the short term if the models do not generalize (though I do think having it as an example of what other groups could and should do alone makes it worthy of acceptance).
> >
> > I have otherwise left my original review as-is, to make it clear what the authors are responding to, but increased the score; I look forward to seeing the hyperparameters added to the supplement. Congratulations and very best of luck!

---

### Official Review · Reviewer_LuiL · 2022-07-28
**A well-documented and useful new benchmark for brain structure analysis**

**Rating:** 7
**Confidence:** 4
**Correctness:** Yes, all good.

**Strengths:**

The dataset and benchmark span multiple scales, from pixel-level microstructure to gross brain region labels (macro structure) within a contiguous tissue sample. There are few machine learning datasets that integrate information across such scales. These benchmarks will be relevant for researchers making sense of the larger and larger structural datasets being generated by serial section electron microscopy.

The paper is well-written, and the dataset and benchmarks are well-documented.

The authors establish 3 exciting benchmark tasks and test a comprehensive set of approaches on each, including both supervised and self-supervised techniques.

The dataset, models, results, and code are all readily available.

**Weaknesses:**

I think annotations were all published previously, together with the dataset. This is fine, but can the authors clarify in the text exactly what their contribution is?

Sorry if I missed it, but it looks like there is no discussion of the benchmark's limitations. Is it an issue that this is only one brain? How can the dataset be improved?

It is not super clear from the paper why having data and annotations from multiple scales in a single dataset is important. What if all of the sub-volumes/pixel-labels had come from different brains/datasets? Couldn't models trained on separate, single-scale datasets achieve similar results on the tasks (even Task 3)?


**Additional Feedback:**

N/A

**Clarity:**

The paper was a pleasure to read!

I am confused, when you say, (*emphasis mine*), "These densely-annotated volumetric cutouts *also* contain pixel-level microstuctural labels," do you mean that the dense annotation is somehow distinct from the pixel-level labels?

**Documentation:**

Yes

**Relation To Prior Work:**

See weakness above -- the authors say they inroduce new annotations, but it is unclear which of the annotations are new with respect to their published dataset paper.

**Summary And Contributions:**

The authors present a clearly-communicated and thorough new set of benchmarks for inferring brain macro- and microstructure from an X-ray microtomograph of a section of mouse brain. Although the dataset is already published, here the authors format and present the dataset, via a new set of benchmarks, as an important asset for image analysis and machine learning.

---

> ### Author Response · Authors · 2022-08-11
> **Response to reviewer LuiL**
>
> We thank the reviewer for the thoughtful comments and feedback. In what follows, we provide (i) responses to your main comments below and (ii) pointers to the revised text where we have revisions added (major changes and added text are highlighted in Red).
>
> ***1. “I think annotations were all published previously, together with the dataset. This is fine, but can the authors clarify in the text exactly what their contribution is?”***
>
> This benchmark paper adds additional pixel-level annotations on the existing dataset both at the micro and macro-structural levels, new features extracted from the fine-scale microstructure segmentations, as well three evaluation tasks at different levels of abstraction. In particular, the original paper provided 11 microstructure-annotated 2D slices for each of the 4 subvolumes corresponding to the interest regions. In this work, we provide full microstructure annotations for each subvolume, therefore adding 349 annotated slices for each of the 4 subvolumes. Furthermore, the original paper provided 8 region-level slices as macrostructural annotations. We have annotated the regions in between these slices, resulting in 342 new region-level slice annotations. We have added Lines 150 - 163 (section 3.1) in the revised paper to convey with more clarity how our current work differs from our previous work.
>
> ***2. “It is not super clear from the paper why having data and annotations from multiple scales in a single dataset is important. What if all of the sub-volumes/pixel-labels had come from different brains/datasets?”***
>
> An individual brain has hierarchical structure at multiple spatial scales, which has traditionally been studied using MRI, microscopy and so forth. As high resolution datasets grow in size, there is a new opportunity for algorithms which can leverage the shared structure which underlies this hierarchical organization. For example, an analysis capable of predicting cell density and brain regions in a single sample may give new insight into how microstructure changes at boundaries between regions. Due to individual and experimental variability, it is unclear if such a nuanced understanding could arise from a dataset collected from different brains and modalities.
>
> ***3. Couldn't models trained on separate, single-scale datasets achieve similar results on the tasks (even Task 3)?”***
>
> While many of the baseline models are indeed trained on a single task, we envision that this benchmark can open up the possibility for multi-task models that can both simultaneously parcellate large brain volumes into areas and also segment microscale structures, and use context from the ROI they are in to improve their segmentation and vice-versa. For Task 3, we really wanted to push the limits of unsupervised (SSL) models and see if we can decode more complex attributes from an image from a discriminative model’s representations (models trained in T1).
>
> ***4. “I am confused, when you say, (emphasis mine), "These densely-annotated volumetric cutouts also contain pixel-level microstructural labels," do you mean that the dense annotation is somehow distinct from the pixel-level labels?”***
>
> By ‘dense annotations’, we mean we now have continuous volumes of annotated data (4 volumes, one from each region) with both region-level and microstructural labels, where in our previous work we had ‘sparsely’ annotated slices that were isolated from each other (with respect to continuity of annotations across the slices). We have made it clearer in our revised paper in Lines 150-163.
>
> Thanks again for your detailed review and feedback! Let us know if you have further questions.

---

### Official Review · Reviewer_tvd1 · 2022-07-29
**A grounded multi-task challenge that would benefit from more detail**

**Rating:** 5
**Confidence:** 4
**Correctness:** The evaluation methods are appropriate.

**Strengths:**

The paper is well-written and clearly motivated by the need for a multiscale brain structure challenge. In addition, it is well structured and generally detailed (see weaknesses for exceptions), making it easy to read. Concisely the strengths are:
- A clear motivation in that a benchmark driven by multiscale representations are not common
- The benchmark tasks are well-defined and intuitive (area, region classification, pixel classification)
- A decent number of benchmarks covering both convolutional and non-convolutional methods

**Weaknesses:**

Although the paper has many strengths as described earlier, it is held back by some limitations. Some of these issues are easily fixable, others require significant work.
- The primary weakness is the small sample size. From what I can gather, the dataset contains 48 3D volumes from a single animal. This limits the potential models to those that work well in low-data settings and excludes the application of sample-inefficient methods, such as many self-attention models.
- Not only do there seem to be few samples but the number of samples are not apparent from the main text. It is not clear in the supplementary either. More obvious and detailed descriptions of the train and test splits. The number of images and train/test splits are included in the code configuration files, where I got these numbers. Authors, please correct me if this is incorrect, but these should be provided front and centre in the main text.
- Another weakness is the lack of discussion on the limitations of the work. Surely the authors can identify some limitations worth acknowledging and discussing, these could include: sample size, label assignment, baseline training, potential biases etc.

**Additional Feedback:**

- Please check abbreviation definitions, including SSL and ROI. Are they defined on first use and then the abbreviation used thereafter?
- "Accompanying these new tools for data generation have been major advances in machine learning" this sentence might need rephrasing or a comma to improve clarity.
- Add UK Biobank to existing datasets
- What is the number of samples included in the dataset? Is it the same animal subject in both?
- I recommend using 8-bit unsigned integer instead of uint8
- Table 1: is this binary classification? 4 classes? What would a naive or random model achieve?
- Table 2 and 3 caption: What performance metric is being used here?
- Table 3: There is an extra space after the decimal for one of the BYOL results


**Clarity:**

In general, the paper is well written, however it is not perfect. Aside from the missing data details mentioned earlier there is missing clarity in the table figures regarding metrics and tasks (see questions to authors for more details).



**Documentation:**

As a benchmark paper, the code repository has sufficient details to run experiments to compare on the benchmark.

However, there are details lacking in the paper. For example, what is the train/test split? How many samples in each? How was the split decided? What are the frequencies of the different classes? Etc.

**Ethics:**

The data collection and animal experiment approvals are covered in the original data release: Prasad, J.A., Balwani, A.H., Johnson, E.C. et al. A three-dimensional thalamocortical dataset for characterizing brain heterogeneity. Sci Data 7, 358 (2020). https://doi.org/10.1038/s41597-020-00692-y

I have no reason to dispute those ethics approvals.

Similarly, I see no ethical concerns surrounding the benchmark tasks presented here.

**Relation To Prior Work:**

The related work discussion is adequate but would benefit from an additional reference to UK Biobank, which is currently the largest open resource of brain MRI scans and is routinely used in ML research.

**Summary And Contributions:**

The paper presents a benchmark challenge on mouse brain MRI data. The three tasks consist of macro-level region classification, pixel-level microstructure classification and estimation of semantic features.

The three tasks cover a wide range of scale from pixel level to whole image regions. Previous challenged have mostly featured a single scale, e.g. whole-image classification or single scale segmentation. The set-up presented by the authors allows for models to learn on the same data across scales.

The dataset is available (released in previous publication), the contribution here is the creation of a benchmark challenge with available code and baseline models for comparison.

---

> ### Author Response · Authors · 2022-08-11
> **Response to reviewer tvd1 (1/2)**
>
> We thank the reviewer for the thoughtful comments and feedback. In what follows, we provide (i) responses to your main comments below and (ii) pointers to the revised text where we have revisions added (major changes and added text are highlighted in Red).
>
> ***1. “The primary weakness is the small sample size. From what I can gather, the dataset contains 48 3D volumes from a single animal. This limits the potential models to those that work well in low-data settings and excludes the application of sample-inefficient methods, such as many self-attention models.”***
>
> The full dataset that we build our benchmark on is a microCT generated 3D brain volume of dimensions 5805x1420x720 (x,y,z) with an isotropic voxel volume of 1.17um, and spans a large intact section from hypothalamus to the cortex. We make this data and new annotations available through the BossDB platform for accessibility, visualization, and easy-to-use APIs for pulling down and uploading data and annotations. From this full data, we select 4 subvolumes of size 256x256x360 (task ROI-C1 and task 2), one each from Cortex, Striatum, VP and Zona Incerta (10-frame buffer between train/val and test sets to be excluded). This yields 1,400 image samples for 2D tasks (this number quadruples to 5600 for ROI-C2 and ROI-C3) and 348 volume samples for 3D tasks (assuming 4 slices per volume, which yields the best performance). For learning on these samples from all the 4 subvolumes, we create and make available high-quality human-verified dense annotations.
>
> Though we collect a significant amount of samples, we acknowledge that it would still lie in the low-data regime and thus deploy models that can work with limited data. In this field of neuroscience, obtaining such high-quality labeled data (especially for dense segmentation tasks like task 2), is very costly. The intensive nature of creating and proof-reading manual annotations limits us from being able to make more labeled and annotated data available. Though this is a limitation, it also is an accurate reflection of the challenges faced in the field, and hence by bringing forward this benchmark and making it easily accessible, we wish to encourage development of more label-efficient approaches for learning. Also, the data we provide is from a single animal subject as our keen interest was to focus on enabling models that could learn to operate in diverse regions and scales of interest within a single subject’s brain. In the revision, we provide new text to address these points in Lines 356-374.
>
> ***2. “More obvious and detailed descriptions of the train and test splits ... These should be provided front and centre in the main text.” “How many samples in each? How was the split decided? What are the frequencies of the different classes? Etc.”***
>
> We thank the reviewer for pointing out places where the dataset and task specs can be made more clear. We did report the train/test splits in the original submission but think that some of these details could be lost due to the fact that the training and testing vary across the different tasks. To make it more clear for readers, we have incorporated this information for the different tasks in lines 216, 221, 227 and 234 of the revised paper. Also, similarly we have added information regarding the class frequencies in Appendix Section 5.2.3 in the revised paper.
>
> In regards to your question about our rationale, we split our data in Task 2 to make approximately 80% of the data available to train on and then a small buffer to avoid any similarity between train and test slices. The buffer size was decided based on a correlation analysis between all of the images, which is detailed in Section 1.1 of the Appendix. In Task 2 and the first subtask of Task 1 (ROI-C1), we follow this rationale and use the same overall train-val-test split for all subtasks. In the evaluation of the generalization of our classification models in Task 1, we also extracted 12 new volumes (3 per interest region) -- and we did this based on a spatial exploration of the new ROI interpolations (Appendix sec 1.3), trying to keep them as distanced from each other as possible to promote heterogeneity.
>
> ***3. “The related work discussion is adequate but would benefit from an additional reference to UK Biobank, which is currently the largest open resource of brain MRI scans and is routinely used in ML research.”***
>
> We thank the reviewer for their suggestion and will have included this reference to our discussion of related work in the revised paper (Line 102). We do note that the bulk of our related work section was focused on high-resolution data that is more similar to our current study (X-ray is 1.17 micron in 3D, whereas conventional MRI is closer to 0.5-1.5 mm).
>
> (continued...)

---

> > ### Author Response · Authors · 2022-08-11
> > **Response to reviewer tvd1 (2/2)**
> >
> > **4. Additional Feedback**
> >
> > Thank you for your additional feedback and helpful suggestions. Some answers are provided below for those points that we didn’t address above.
> >
> > +
> > **What is the number of samples included in the dataset? Is it the same animal subject in both? -- >** We have provided further details on the dataset sizes for the different tasks in Lines 135-163 and have made the single individual nature of the dataset more clear on Line 136.
> > +
> > **I recommend using 8-bit unsigned integer instead of uint8  -- >** Occurences of the uint acronym has been replaced with expanded form - ‘unsigned integer’ as recommended.
> > +
> > **Table 1: is this binary classification? 4 classes? What would a naive or random model achieve? -- >** In lines 186-195 (section 3.2, para 3), we had originally specified the classes involved in each task, further in the revised paper we have added the number of classes to the tables and captions. We provide further details on the ROI classification problem in Section 3.3.1.
> > +
> > **Table 2 and 3 caption: What performance metric is being used here? -- >** Regarding the performance metrics used in the tables, for task 2, we do specify them as F1 and IoU in the table itself and also it is detailed in lines 298-299 (Section 4.2, para 2). In order to make it more obvious, we have added a separate ‘metric’ column for Table 2 in the revised paper. For Task 1, we report classification accuracy as the main metric.
> > +
> > **Table 3: There is an extra space after the decimal for one of the BYOL results” -- >** Corrected
> >
> > Thanks again for your detailed review and feedback! Let us know if you have further questions.

---

### Author Response · Authors · 2022-08-11
**Thank you for your comments and feedback (1/3)**

First, we would like to thank the reviewers for their valuable comments and feedback on our work. We are honored to receive good feedback from reviewers: tvd1 points out that our work is “clearly motivated”, and Reviewers tvd1 and LuiL describe the paper as “well-written”. Furthermore, we deeply value the collective recognition from reviewers tvd1, LuiL, ncab and RzzT that “benchmark driven by multiscale representations are not common” and that “there are few machine learning datasets that integrate information across such scales''. We also appreciate the validation by reviewers LuiL, JzxN and aUux that our code is “well-documented” and the recognition from reviewer Rzzt that we have “gone extra miles to offer usability convenience for computer scientists”.

Based on the feedback from reviewers, we were able to identify a number of areas for improvement, such as the clarity of the text and explanation of the dataset, our contributions over our previous dataset release (which now includes fully 3D volumetric dense annotations of microstructure), and adding a discussion on the limitations of the work. We have already started to incorporate some of these revisions to the paper and highlight significant changes and additions to the text in Red. We will continue to work on reviewer suggestions through the rebuttal period.

In the rest of this general response, we will respond to some of the key questions that were raised by reviewers and provide further context about the significance and contributions of this paper and benchmark.

**1. Why multi-task and multi-scale analysis matters for brain mapping**

For this benchmark, our goal was to develop ideas and tools for neuroanatomical multi-scale, multi-task analysis. The tasks we prepared are designed to exploit the high resolution (1 micron) across-area span of our dataset that makes it possible to assimilate the rich heterogeneity of the brain structures, which is rarely possible due to the difficulty of collecting and curating neuroanatomical data at such resolution and span. However, it is anticipated that increasing numbers of high-resolution, large-scale datasets will be collected in the near future driving the need for such analysis tools.
As higher resolution brain data becomes available, there is a new opportunity for algorithms which can leverage the shared structure which underlies this hierarchical organization. This is a distinct advantage of this data- it is possible to exploit and analyze the relationship between structure at different levels to improve performance but also to provide insight into the nuanced relationship between these levels of abstraction (for example, how cell density and structure varies at the boundaries of traditionally understood brain regions). Having multi-scale tasks on a single dataset can also enable the study of how relevant specific features are at different scales or tasks without introducing additional variability due to differences in sample preparation and the specifics of the different samples imaged in different experiments. Achieving this level of understanding with the variability of multi-modal and multi-sample imaging on different individuals may be quite challenging due to individual sample variability.

**2. Dataset Size and Generalization**

**Isolating the effects of brain heterogeneity from across-sample imaging parameters and sample preparation:**  When designing this benchmark, we decided to focus on expanding the collection of tasks and types of models (both supervised and self-supervised) which enables learning across scales the different structures inherent within the same brain. As we introduce new datasets or animals, it becomes more difficult to model the inter-area variability within the structure and organization of the brain due to variability in sample preparation and differences in imaging conditions.
Based upon the reviewer feedback, we understand that the data sample being from a single brain is felt as a limitation of our current work. While we agree that being able to expand this dataset to new animals would be of great significance, generating the datasets, tasks, and expanded annotations is also no easy task. We hope to expand our efforts to new modalities and samples where multi-scale annotations and images are also provided.
In our revised paper, we will discuss the challenges in generalization when collecting data at this scale, and how including more animals and preparations in the future could be an interesting line of future research.

(continued...)

---

> ### Author Response · Authors · 2022-08-11
> **Thank you for your comments and feedback (2/3)**
>
> **Clarification of dataset size:**  The dataset that we build our benchmark on is just a subset of an even larger, intact volume of the mouse brain that spans many brain regions. We point out that because most existing micron-scale X-ray datasets have focused on a single area at a time [1,4] this dataset is actually much larger than most intact brain datasets at this spatial resolution and extent. Generating these datasets is still challenging; however, with the model results generated through this effort, the field can improve and build more unified data generation approaches in the future.
>
> Detailing the specifications, the dataset that we build our benchmark on is a contiguous volume amounting to ~5,935M voxels spanning 8 potential ROIs. The resolution is 1 micron isotropic, which means that in 3D, we can get micron-resolution information from the intact brain and resolve axons (white matter), cell bodies, and vasculature in great detail. It is from this original volume that we significantly expand our previous sparse annotations to now provide full 3D pixel-level labeled data across 4 different subvolumes within the dataset of particular anatomical importance, each of which are of ~23.6M voxels. The collection and curation of this type of labeled data becomes very intensive as it spans multiple regions without sacrificing on resolution where other high-resolution 3D imaging data are often constrained to a small portion of a single brain region [1] or otherwise when the span may be large, the resolution tends to be much lower [2]. In addition, in Task 1, we build ROI-level challenges across an even wider extent of the full brain volume -- where it is possible to easily extend our ROI-level annotations to even larger swaths of the full dataset and test generalization.
>
> As such, we see the ideas and tools that we have developed for neuroanatomical multi-scale, multi-task analysis to be an important contribution of the work, and this general framework could be used in other imaging modalities like in 2D pathology images of large multi-organ or multi-tissue sections in the future. We hope to expand this effort in future benchmarks and challenges.
> Nevertheless, we want to acknowledge the generalization and sample size limitations of our current dataset and have included a discussion on these limitations. Please see lines 356-374 for an expanded discussion in our revised paper.
>
> **3. Discussion of limitations, maintenance and ethical use**
>
> Reviewers tvd1 and LuiL point out the lack of a discussion on limitations of the work. We agree that this is needed to provide readers with insights into what the current issues and challenges are, and how to improve the dataset and related benchmark efforts in the future. We now acknowledge the different limitations raised by the reviewers (e.g., dataset size, only a single individual in data) and have started our revision to include a detailed discussion in lines 356 to 374.
>
> We are thankful to reviewer ncab for pointing out that a discussion on maintenance and ethical use is lacking. We acknowledge that these key points were not adequately addressed and we have incorporated a separate section discussing in detail how the dataset, API and codebase will be maintained, how community contributions will be managed, and the ethical protocols and considerations for this dataset. This discussion has been added in lines 671-684 of the appendix.
>
> **4. Contributions of the work to building infrastructure and models for BossDB**
>
> One point we want to raise about our current effort, which we think unfortunately we did not communicate clearly, is that we have built our examples and model deployment etc on BossDB, which is the BRAIN Initiative archive of record for nanoscale connectomics datasets and houses a growing number of other related X-ray and EM datasets. While they don’t yet have the same level of dense annotations of multi-scale structure, because of which we didn’t include them in our current work, our benchmark effort also will serve to accelerate development of machine learning tools for semantic segmentation and analysis of other datasets in BossDB. This includes datasets from different: 1) species [4], 2) modalities (eg.: Electron Microscopy [5]), 3) experimental conditions (eg.: developmental stages [6]). We intend to host future challenges built on new datasets using the models and task definitions established by this dataset. To emphasize this point, we have added a brief paragraph across Lines 196 - 204 (Section 3.2) and have highlighted this in the Discussion section (Lines 351-355).
>
> **5. Significance of mouse models in expanding our understanding of the brain**
>
> Reviewer aUux points out that the dataset may have limited generalization or impact because it has been collected in the mouse brain.
>
> (continued...)

---

> > ### Author Response · Authors · 2022-08-11
> > **Thank you for your comments and feedback (3/3)**
> >
> > However, we want to remind the reviewers that not only do the mouse and human brain share a lot of similarities in their architecture, organization, and cell types, ongoing US BRAIN Initiative programs are focused on whole-brain mouse connectivity mapping as a major scientific step forward in understanding mammalian brains [3]. Tools for processing and analysis of mouse data remain critically relevant to the overall neuroscience community.
> >
> > In addition, several key discoveries have been derived first in mouse models before discovery in other mammalian systems. The resolution of the data we provide here and the potential for multi-resolution analyses of 3D data provided by our benchmark models can be helpful in imaging studies at different spatial scales, like EM (at higher res) and MR to visualize water diffusion at coarser resolutions [7].
> >
> > ___
> > References:
> >
> > [1] Dyer, Eva L. et al. "Quantifying Mesoscale Neuroanatomy Using X-Ray Microtomography". eNeuro 4. 5(2017).
> >
> > [2] Griffin Rodgers, et al. "Virtual histology of an entire mouse brain from formalin fixation to paraffin embedding. Part 1: Data acquisition, anatomical feature segmentation, tracking global volume and density changes". Journal of Neuroscience Methods 364. (2021): 109354
> >
> > [3] National Institutes of Health . (n.d.). RFA-NS-22-048: BRAIN initiative connectivity across scales (BRAIN CONNECTS): Comprehensive centers for mouse brain (UM1 clinical trial not allowed). Retrieved August 11, 2022, from https://grants.nih.gov/grants/guide/rfa-files/RFA-NS-22-048.html
> >
> > [4] Kuan, A.T., Phelps, J.S., Thomas, L.A. et al. Dense neuronal reconstruction through X-ray holographic nano-tomography. Nat Neurosci 23, 1637–1643 (2020). https://doi.org/10.1038/s41593-020-0704-9.
> >
> > [5] Kasthuri N, Hayworth KJ, Berger DR, Schalek RL, Conchello JA, Knowles-Barley S, Lee D, Vázquez-Reina A, Kaynig V, Jones TR, Roberts M, Morgan JL, Tapia JC, Seung HS, Roncal WG, Vogelstein JT, Burns R, Sussman DL, Priebe CE, Pfister H, Lichtman JW. Saturated Reconstruction of a Volume of Neocortex. Cell. 2015 Jul 30;162(3):648-61. doi: 10.1016/j.cell.2015.06.054. PMID: 26232230.
> >
> > [6] Witvliet, D., Mulcahy, B., Mitchell, J.K. et al. Connectomes across development reveal principles of brain maturation. Nature 596, 257–261 (2021). https://doi.org/10.1038/s41586-021-03778-8.
> >
> > [7] Foxley S, Sampathkumar V, De Andrade V, Trinkle S, Sorokina A, Norwood K, La Riviere P, Kasthuri N. Multi-modal imaging of a single mouse brain over five orders of magnitude of resolution. Neuroimage. 2021 Sep;238:118250. doi: 10.1016/j.neuroimage.2021.118250. Epub 2021 Jun 9. PMID: 34116154; PMCID: PMC8388011.
> > ___

---

### Meta-Review · Area_Chair_TD98 · 2022-09-12

**Recommendation:** Accept
**Confidence:** 4

**Metareview:**

This paper introduces an MTNeuro benchmark on a large 3D X-ray microtomography imaging dataset that encompassed multiple brain areas. Three tasks are designed covering a wide range of scale from pixel level to the whole image regions, and various supervised as well as self-supervised algorithms, are evaluated on the established benchmark tasks.

In general, the paper is clearly motivated and well-structured. The created dataset is interesting, which may benefit the research in this area. However, the proposed dataset is quite small in size limiting its applicability and generalizability in real-world scenarios. In addition, the paper should be self-included, instead of directing the readers to their portal for details. In summary, I vote for weak acceptance of this paper, although would not be upset if it were rejected.

---

### Decision · Program_Chairs · 2022-09-16

Accept